# A Comparison of Wear Patterns on Retrieved and Simulator-Tested Total Knee Replacements

**DOI:** 10.3390/jfb13040256

**Published:** 2022-11-19

**Authors:** Rebecca H. Dammer, Carmen Zietz, Rainer Bader

**Affiliations:** Biomechanics and Implant Technology Research Laboratory, Department of Orthopaedics, Rostock University Medical Center, Doberaner Strasse 142, 18057 Rostock, Germany

**Keywords:** total knee replacement, retrieval analysis, wear pattern, wear simulator study

## Abstract

Aseptic implant loosening is the most common reason for revision surgery after total knee replacement. This is associated with adverse biological reactions to wear debris from the articulating implant components. To predict the amount of wear debris generated in situ, standard wear testing of total knee replacement (TKR) is carried out before its clinical use. However, wear data reported on retrievals of total knee replacement (TKR) revealed significant discrepancies compared with standard wear simulator studies. Therefore, the aim of the present study was to compare the wear patterns on identical posterior-cruciate-retaining TKR designs by analyzing retrieved and experimentally tested implants. The identification and classification of wear patterns were performed using 21 retrieved ultra-high-molecular-weight-polyethylene (UHMW-PE) inserts and four sets of inserts of identical design and material tested in a knee wear simulator. These four sets had undergone different worst-case conditions and a standard test in a wear simulator according to ISO 14243-1. Macroscopic and microscopic examinations of the polyethylene inserts were performed, including the determination of seven modes of wear that correspond to specific wear patterns, the calculation of wear areas, and the classification of the damage over the whole articulating area. Retrieved and standard wear simulator-tested UHMW-PE inserts showed significant differences in wear area and patterns. The total wear areas and the damage score were significantly larger on the retrievals (52.3% versus 23.9%, 32.7 versus 22.7). Furthermore, the range of wear patterns found on the retrievals was not reproducible in the simulator-tested inserts. However, good correspondence was found with the simulator-tested polyethylene inserts under worst-case conditions (third body wear), i.e., deep wear areas could be replicated according to the in vivo situation compared with other wear test scenarios. Based on the findings presented here, standard simulator testing can be used to directly compare different TKR designs but is limited in the prediction of their in situ wear. Preclinical wear testing may be adjusted by worst-case conditions to improve the prediction of in situ performance of total knee implants in the future.

## 1. Introduction

Total hip and knee replacement has become a very common and successful surgery. Endoprosthetic registers show increasing numbers of implantations in recent decades, accompanied by increasing numbers of revision surgeries [1,2]. The survival rate for total knee endoprostheses is given by Civinini et al. [3], about 85% after 13 years for younger and about 95% for older patients. However, the lifetime of total knee replacement (TKR) depends on several factors. Implant-related factors, for example: level of constraint, design, material combination, and size matching, as well as patient-related factors such as: age, activity level, and body mass index play a decisive role. Revisions are cost-intensive, more invasive than primary surgery, and associated with twice the risk of infection [4]. Aseptic implant loosening is the most common reason for revision surgery; these can cause severe osteolysis, resulting in periprosthetic bone defects [5,6,7,8,9].

As the main reason for the aseptic loosening of endoprostheses, osteolysis due to adverse biological reaction to wear debris is known [3,10,11]. It is established that wear debris may also trigger implant-related infections and stimulate septic implant loosening [5]. In hard-on-soft bearings, the polymeric articulating partner is considered the weakest component as it has lower wear resistance when compared with the metallic or ceramic articulation partner. Thus, the TKR lifetime is mainly defined by the wear performance of the weakest implant component.

Before endoprosthetic implants are introduced for clinical use, they must pass through several investigations. These preclinical tests should check for the intended functions in situ. For total joint replacements, in particular, the lifetime or durability is approximated by analyzing the wear behavior on the human body. The wear simulation of total hip and knee replacements according to ISO 14242 [12,13] and ISO 14243 [14,15] provides data for approximating the average wear rate of the implants in situ.

However, inaccurate preclinical prediction may lead to underestimated wear rates and overestimated implant lifetime. This could result in false assumptions for specific implant designs that might be excessively retrieved in the following years. For example, the Australian Orthopaedic Association National Joint Replacement Registry provides an annual report in which revised total knee endoprostheses with a higher than anticipated revision rate are mentioned [16].

The analysis of retrieved and simulator-tested polyethylene inserts of identical designs can facilitate the direct comparison of the amount and patterns of wear between in vivo and experimental wear simulation. For total hip replacement, it is stated that standard wear simulator tests cannot predict the in vivo situation in an exact manner for the entire time of the implantation [11,17,18]. Additionally, for total knee replacement (TKR), some studies of retrievals and standard wear testing showed a significant discrepancy in wear data [19,20,21,22,23]. Previous studies had focused on implant malpositioning, overloading, and reasons for enhanced wear of TKR [21,24,25]. Some studies aimed to identify, detect, and quantify specific wear patterns of TKR [7,25], but the researchers used different implant designs to compare retrievals with implants from wear simulator studies or different computational approaches or compared the wear without examining the specific wear pattern [19,20,21,23,26]. Harman et al. and Paltanea et al. [21,27] showed the potential of the use of optical microscopy techniques in the examination of failed implants. They used grading systems that provide the opportunity to compare cross-design. These studies pointed out discrepancies in wear patterns between in vivo and experimental simulation.

To the best of the authors’ knowledge, none of the previous studies compared identical TKR designs of retrievals and simulator-tested implants. Therefore, our study was aimed at comparing the wear patterns on identical posterior-cruciate-retaining TKR designs by analyzing retrieved and experimentally tested implants in standard and severe test conditions. Retrieved ultra-high-molecular-weight-polyethylene (UHMW-PE) inserts of identical implant design and material (in total 21) were examined. Furthermore, experimental simulator tests were performed, including testing according to ISO standards and different worst-case scenarios such as third body wear (TBW) and malpositioning of the implant components (on additional 12 inserts). To determine wear behavior, the area and patterns of wear of the UHMW-PE inserts were quantified. Direct comparison between retrieved inserts and simulator inserts tested with different worst-case test scenarios provided the chance to identify potential causes of excessive wear and early implant failure.

## 2. Materials and Methods

### 2.1. Implant Components

A bicondylar cruciate-retaining TKR system (Multigen Plus, Lima Corporation, San Daniele, Italy) was used for the simulator tests and retrieval analysis. The tibial component of the system included a tibial slope of 6°. The UHMW-PE inserts were made of GUR 1050. The molecular weight of the inserts was between 3 and 6 million g/mol [28]. The inserts were sterilized using ethylene oxide. All the inserts were stored dry and dark until examination.

### 2.2. Wear Simulator Study

Wear simulator studies of the Multigen Plus CR knee system were performed using a servo-hydraulic knee simulator (C3/1-08; EndoLab GmbH, Rosenheim, Germany), according to ISO 14243-1 [14]. In all investigations, size 3 of the femoral components, tibial components, and UHMW-PE inserts were used. The simulator consisted of three running stations and one loaded station that acted as soak control. In addition to the standard test according to ISO 14243-1 for five million cycles, worst-case tests were realized for 2.5 million cycles. The procedures used for the standard and worst-case simulator tests are described in detail by Zietz et al. [22,29,30]. The worst-case scenarios included an internal rotation (IR) of the femoral component of 9° and a posterior tibial slope (TS) of the insert of 10°. TBW caused by bone cement particles was also simulated. The materials of the femoral components are listed in Table 1. Simulated lifetime was calculated for comparing the performed cycles of the simulator-tested inserts and the retrievals, postulating that a patient supported with TKR would perform 2.02 million cycles/year according to Battenberg et al. [31].

### 2.3. Retrieval Analysis

The implant retrievals (cemented, bicondylar cruciate-retaining TKR systems manufactured by Multigen Plus, Lima Corporation, San Daniele, Italy) were derived from revision surgeries at the Department of Orthopaedics at the Rostock University Medical Center. The retrievals were disinfected and subsequently stored postoperatively. The examinations were reported and approved by the ethics committee of the University of Rostock (Register number: A 2016-0187).

A total of 21 retrieved UHMW-PE inserts (9 from the right side and 12 from the left side) with a lifetime in situ ranging between 0.2 and 6.5 years ((1.9 ± 1.9) years) were used for the analysis. The sides were normalized by transferring the compartments into medial and lateral parts. Insert sizes ranged from 1 to 4, with most inserts (*n* = 12) being size 3. In seven cases, the size of the femoral component differed from the size of the insert. In each case, the femoral component was one size larger than the UHMW-PE insert. In five cases, the articulating partner was a ceramic component (Biolox Delta^®^). In all other cases, the metallic femoral component was made of Co28Cr6Mo. The thickness of the inserts ranged from 10 mm (*n* = 13) to 17 mm (*n* = 1). The storage time before examination ranged from 2 to 10.5 years for the retrievals and 1 to 2 years for the inserts from simulator tests.

### 2.4. Examination of Wear Area and Patterns

#### 2.4.1. Wear Area

Previous studies [7,19,21,24,25,26,27] established that wear areas and patterns on the retrievals and implants from wear simulator studies could be reliably detected and classified. The wear patterns on the articulating surface of all the inserts were signed under changing light using water-soluble markers without magnification. Differentiation was made between unclear wear (presumably damaged during revision, not mentioned further), total wear (every visible wear mark, including scratches on the remaining manufacturing marks), and deep wear (wear that completely rubs off the manufacturing marks). The total wear area minus the deep wear area gives the superficial wear area.

Overview pictures were taken using the same scale and under the same conditions (Canon EOS 600D, lens: Macrolens, EF 100 mm, 1:2.8, USM). To ensure the same distance from the proximal surface to the lens, all the implants were lifted to the height of the thickest implant (h = 17 mm). Pictures were subsequently loaded into an image processing program (Leica QWin V3, Leica Microsystems, Wetzlar, Germany), and areas of the wear patterns were measured in two dimensions. The areas of the proximal surfaces were calculated as projected surfaces in two dimensions using CAD software (SOLIDWORKS ^®^, Dassault Systèmes, Vélizy-Villacoublay, France) for all insert sizes. Subsequently, two-dimensional wear areas were normalized based on the inserts size as Harman et al. suggested [21]. Mediolateral and anteroposterior elongation of the wear area were measured using a digital microscope (VHX-6000, Keyence, Osaka, Japan) with magnifications of 5×. These values were normalized to that of an insert of size 3. For localization of the wear scar, the anteroposterior elongation was measured from the centerline (Figure 1) in both directions. The ratio of the anterior and posterior parts gives a value for the location of the wear scar in the anterior and posterior compartments of the insert. Values between 0 and 1 indicate a more posterior localization of the wear scar, while values more than 1 indicate a shift of the wear scar to the anterior. Possible backside wear was not analyzed in the present study.

#### 2.4.2. Wear Patterns

Following Wasilewski et al. [25], the proximal surface of the inserts was dissected in ten areas (Figure 1). The articulating surface was examined microscopically with magnifications of 10×. Based on similar observations in literature [7,24,25,26,29,32,33,34], the following patterns of wear were identified and related to each of the ten areas in Figure 2.

The characteristics of articulation surfaces, such as delamination and embedded debris, were not found on any of the UHMW-PE inserts.

As Hood et al. suggested [7], a subjective grading system was used to calculate the damage score for each of the inserts. Instead of a percental approximation of the areas of each wear pattern, the wear patterns in the compartments (Figure 1) were counted based on the severity and summarized to a damage score. The grades assigned were: 0—not present, 0.5—present but not eminently, and 1—severely present. Severe presence might indicate the presence of the wear pattern on more than 50% of the compartment and an eminently distinct presence on a relatively small area of the compartment. The study intended to compare retrieved and wear simulator-tested inserts; therefore, the subjective component of the damage score was not tested for reliability.

### 2.5. Statistical Analysis

Statistical significances and correlations were calculated using IBM^®^ SPSS^®^ Statistics Version 27 (IBM, Armonk, New York, NY, USA). The parameters that were evaluated in this study were the abovementioned pattern of wear (burnishing, abrasion, pitting, scratching, striated wear pattern, TBW, deformation, and rim-runner) and wear area (medial and lateral percentual area of total, deep, and superficial wear, dimension of medial and lateral area of wear in antero-posterior and medio-lateral direction, and localization of wear scar in antero-posterior direction) as well as implant-related factors (lifetime, size of the femoral component, size of the insert, and height of the insert).

Significant differences between retrievals and simulator-tested groups were calculated using the Mann-Whitney U test. Correlations between the examined characteristics were calculated using the gradual coefficient of correlation according to Kendall (Kendall-Tau-b) for the group of retrievals. Only strong correlations will be mentioned. Differences and correlations were marked with one asterisk if they were significant (*p* < 0.05) and two asterisks if they were highly significant (*p* < 0.01). Data were presented as mean ± standard deviation.

## 3. Results

An overview of all the collected data is given in Table 2. The size of the femoral component and size of the insert showed no strong correlations with any of the examined characteristics. The parameter lifetime showed a strong correlation with deep wear area on the lateral compartment of the inserts (τb = 0.581 **). The height of the inserts showed a strong correlation with the occurrence of rim-runners on the medial compartment (τb = 0.500 **) and the total wear area on the lateral compartment of the inserts (τb = 0.504 **).

No significant differences were found between the time in situ of the retrievals (mean: 1.9 years, range: (0.2–6.5) years) and the calculated time in situ of the group of simulator-tested inserts (mean: 2.1 years, range: (1.2–2.5) years).

### 3.1. Wear Areas (in %)

Wear in partitions A and P (Figure 1, middle of the inserts) was found on only two of the retrievals and will not be considered further. The total wear areas (Figure 3) of the retrievals were significantly larger than on the simulator-tested inserts for the CoCr-ISO set (medial: *p* = 0.023, lateral and entire: *p* = 0.001), the TiN-IR set (medial: *p* = 0.011, lateral: *p* = 0.001, entire: *p* = 0.002) the TiN-TS set (lateral: *p* = 0.001, entire: *p* = 0.023), and the entire group of worst-case tested inserts (medial: *p* = 0.004, lateral and entire: *p* < 0.001).

The retrievals’ deep wear areas (Figure 4) were significantly smaller than in the entire group of worst-case inserts (medial: *p* = 0.045, lateral: *p* = 0.028, entire: *p* = 0.022).

The retrievals’ superficial wear areas (difference between total wear area and deep wear area) were significantly larger than on the simulator-tested inserts for all groups (medial, lateral, entire: *p* = 0.001).

### 3.2. Dimension of Wear Areas (in mm)

The anterior–posterior (ap-) elongation of the total wear areas (Figure 5) showed significantly larger values for the retrievals on both compartments when compared with the CoCr-ISO set (medial: *p* = 0.002, lateral: *p* = 0.001), the TiN-IR set (medial: *p* = 0.002, lateral: *p* = 0.007), and the worst-case simulation (medial: *p* = 0.028, lateral: *p* < 0.001) and on the lateral compartment compared with the TiN-TS set (*p* = 0.001).

The ap-elongation of the deep wear areas showed significantly larger values for the retrievals on the lateral compartment when compared with the CoCr-ISO set (*p* = 0.026) and on the medial compartment compared with the TiN-TS set (*p* = 0.026) and the entire set of worst-case simulations (*p* = 0.044).

The medial–lateral (ml-) elongation of the total wear areas (Figure 6) showed larger values for the retrievals on both compartments when compared with the TiN-IR set (medial and lateral: *p* = 0.023) and the set of worst-case simulations (medial: *p* = 0.007, lateral: *p* = 0.008) and on the lateral compartment when compared with CoCr-ISO set (*p* = 0.001) and the TiN-TS set (*p* = 0.004).

The ml-elongation of the deep wear areas did not differ significantly between the retrievals and the simulator-tested groups.

A separation of the ap-elongation of the wear area in a part anterior and a part posterior of the centerline (Figure 1) was performed. The ratio of both parts (anterior/posterior) was calculated for both compartments (Figure 7).

The total wear area on both compartments of the retrievals was for all except three inserts located more to the anterior (ratio > 1). Significant differences compared with the retrievals were calculated for the localization of the medial compartments (TiN-IR: *p* = 0.004, TiN-TS and Worst-case: *p* = 0.007) and the lateral compartments of the CoCr-ISO set (*p* = 0.001) and the TiN-TS test condition (*p* = 0.001). In the CoCr-ISO set, the total wear area on the lateral compartment was located completely posterior to the centerline in the PLM and PLL zones.

The deep wear area on the medial compartments of the retrievals was for all except five inserts located more to the anterior. Significant differences compared with the retrievals were calculated for the localization of the lateral compartments for the CoCr-ISO set (*p* = 0.003) and the TiN-TS set (*p* = 0.005). In the CoCr-ISO set, the deep wear area on the lateral compartment was located completely posterior to the centerline, in the PLM and PLL zones.

The ratios (Figure 7) indicated a rotation of the wear areas around the center of the insert. A slightly external rotated femoral component (or a somewhat internal rotated tibia) could be demonstrated for the retrievals: the medial wear areas were located more to the anterior than the lateral wear areas. This was also found in the CoCr-ISO and the TiN-TS test conditions; however, the loading center was more anterior for the retrievals. The ratio of the TiN-IR set was representative of the tested condition: an internally rotated femoral component (or an externally rotated tibia). For the TBW test conditions, no rotation was visible.

### 3.3. Microscopic Examination

Except for delamination and embedded debris, which were not found on any inserts, each of the remaining eight wear patterns was found on the retrievals. Some wear patterns did not occur on the simulator-tested inserts (Figure 8). No rim-runner, no deformation, no pitting, and only slight third-body wear were found on the inserts in the CoCr-ISO simulation. On the TiN-IR and TiN-TS inserts, six of the eight wear patterns were found. On the CoCr-TBW inserts, seven of the eight wear patterns were found.

## 4. Discussion

The present study aimed to compare the wear in retrieved and simulator-tested UHMW-PE inserts of identical implant design and material to identify potential causes of excessive wear and early failure of TKR. Wear behavior was determined and quantified by examining wear areas and patterns. The total wear on the retrievals was more pronounced than that on the simulator-tested inserts, whereas the deep wear was smaller or comparable. Depending on the test conditions, different wear patterns occurred on the simulator-tested inserts. The examination of the retrievals provided a more complex picture of the potential patterns of wear, which were not reflected by the preclinical simulator tests of total knee replacements. Hence, standard simulator tests according to ISO may underrepresent TKR wear propagation in situ.

However, it was possible to show that the simulator test’s worst-case conditions were more suitable for reproducing the in vivo situation for the examined design. It was possible to show the reasons for specific patterns of wear in the TKR in worst-case test conditions.

The time in situ of the analyzed retrievals (23 months) was short when compared with the average lifetime of bicondylar TKR in patients. A 10-year survival is specified in the literature, with more than 90% for primary TKR [35,36] and 86.1% for revision implants [37]. However, the mean lifetime of the analyzed retrievals is comparable with the mean running times of the analyzed inserts from the simulator tests, according to Battenberg et al. [31]. Studies on retrievals with average lifetimes comparable with those used in the present study are reported in the literature [20,21,23,25].

In agreement with the literature [20,21,23,25], the retrievals showed larger total wear areas than the simulator-tested inserts. This was most likely caused by higher ranges of motion of the TKR in patients in contrast to defined restricted motion patterns in the wear simulator [20,21]. However, the current study reveals that wear induced by third-body particles can cause similar total wear areas as found in the retrievals, as reported by Wasielewski et al. [25]. The total wear areas were subdivided into deep and superficial wear. The findings revealed that simulator testing overestimated the deep wear areas, due to monotonic loading by the repetition of cycles of level walking. The number of deep wear areas on the retrievals was best matched by those found on the CoCr ISO and TiN-IR simulations. In contrast, none of the simulations showed similar values to the retrievals for the superficial wear area. Due to different overlapping areas of knee motion, the dimensions of total and deep wear areas and the size of superficial wear areas showed discrepancies in the retrievals concerning the frequency and duration of daily activities [38], which could not be reproduced in the wear simulator tests with one defined motion so far [22]. In contrast to Harman et al. and Wasielewski et al. [21,25], the wear areas were not predominantly found on the medial or lateral compartments of the retrievals. Wasielewski et al. [25] found insufficiently corrected varus or valgus deformities due to the predominance of wear in the medial or lateral compartment. In the present study, no information was gained regarding a deformity before explantation; the absence of deformations or the design may be the reason for this discrepancy. For the simulator-tested inserts, the predominance was on the medial compartment in the CoCr-ISO and TiN-TS tests but not visible in the TiN-IR and TBW-tests. Literature is inconsistent regarding the predominance of wear in the simulator-tested inserts [20,21,23], which was most likely caused by the differences in the implants tested.

These findings are supported by the results of the current study, describing the extent of the wear areas, ap-elongation, and ml-elongation of total and deep wear. In agreement with Harman et al. [21], the ap-extent of the wear scars was greater for the worst-case tests in the medial and lateral compartments compared with the CoCr-ISO-test. Retrievals also showed higher variations in ap-elongation and higher mean ap-extent for the total wear areas on the medial and lateral sides compared with the simulator tests. These findings are concurrent with other findings in the literature [20,23]. The variation in the ml-elongation was very small for all the inserts; however, in a congruent knee design without a mismatch in size combination, this value might be less critical and therefore not mentioned in literature comparisons of retrievals and simulator tests.

In contrast to the findings of Wasielewski et al. [25] and Harman et al. [21], the wear scars found in the current study were located more towards the anterior on both the compartments of the retrieved inserts. This may be due to a reversed femoral translation, which may have been caused by the inhibited rollback of the femoral component [39]. The localization of the wear areas on the insert may be due to several factors [39,40,41,42]. The examined TKR design has a posterior tibial slope of 6°. In the case of increased slopes, Pourzal et al. [40] predicted more wear due to increased ap-translation and anterior loading, which corresponds to the presented findings in the current study. The localization of the wear areas in the TBW test condition was similar to that of the retrieved inserts.

The rotation of the wear areas in the retrievals showed higher variations than the simulator-tested inserts. The average ratios of the anterior and posterior extent of the wear areas indicated slight external rotation of the femoral component (internal rotation of the tibial component). This was also found in the CoCr-ISO and TiN-TS tests and corresponded to the findings of Harman et al. [21]. In the TiN-IR test condition, the rotation of the wear areas corresponded to the rotated femoral component. The increased wear in the CoCr-TBW simulation due to the particles may be led to the symmetric wear almost completely over the compartments.

Some wear pattern, such as surface delamination, are difficult to reproduce on simulator-tested inserts [22,43]. In contrast to literature data [21,22,24,25,27,32,33,44], no delamination was found in the examined retrievals. Delamination may be caused, for instance, by the type of polyethylene material and sterilization process, height of the inserts [45], implantation time, and aging of the components [38,46].

Wear particles embedded on the implant surface were not found in the retrieved and simulator-tested inserts. This might be due to the cleaning procedure of the retrieved and simulator-tested inserts and the low magnifications used for the microscopic analysis. In agreement with Wasielewski et al. [25], Zietz et al. [29] found embedded wear particles at higher magnification using SEM (scanning electron microscope) in the simulator-tested inserts used under TBW test conditions.

Wasielewski et al. [25] found TBW as a prevalent wear pattern on two unconstrained knee replacements made of UHMW-PE and carbon-reinforced polyethylene. As described in Fabry et al. [24], scratching and TBW are different modes of wear. Still, it might be challenging to point out scratching as the reason for the other surface properties of the articulation partners from slight TBW without further examination under high magnifications. The wear patterns were determined based on the points listed above, which may lead to an overestimation of scratching.

While burnishing, abrasion, and scratching were the main wear patterns found on the simulator-tested inserts, the retrieved inserts mainly showed scratching, pitting, and TBW. The findings for the simulator-tested inserts were in good correspondence with those of previous studies [20,21] in which burnishing and scratching were found to be the prevalent wear patterns after simulator tests. The degrees of burnishing found in the simulator-tested inserts and the retrievals were similar to that reported in previous studies [20,21]. Direct comparison was not entirely possible owing to different designs and types of total knee endoprosthesis (cruciate retaining vs. posterior-stabilized [20]) and different types and heights of polyethylene. Harman et al. [21] reported pitting on polyethylene inserts after the wear simulation of walking. However, this wear pattern was not present on the examined inserts tested under standard and malpositioned conditions. Due to the roughening of the articulation surfaces and higher stresses, pitting seemed to be supported by TBW [44,46,47].

The striated wear pattern established by Wimmer et al. [9] could be identified on the inserts from standard wear test in agreement with Harman et al. [21] and under malpositioning conditions. However, the striated wear patterns were less distinct on the inserts from the TBW test and the retrievals. This might be due to the differences in the analyzed knee endoprostheses, i.e., implant material and design. Rim-runners as found by Wasielewski et al. [25] (specified as cold-flow) and Fabry et al. [24] (specified as creeping at the edge region) are supposed to be a sign of malrotation, or general malalignment and mismatch of the chosen component sizes. In the wear simulation, rim-runners were associated with implant malpositioning (internal rotation of the femoral component and increased tibial slope). On all of the retrieved inserts that were examined, rim-runner was found. The presence of wear patterns in the retrievals examined beyond the actual bearing surfaces may be associated with flat insert design, a false combination of sizes, or for a specific event (trip, stumble, crash, etc.). Rim-runners can occur in case of higher ranges of motion and possible malpositioning of the implants, as the ”height of the insert” parameter revealed medium correlation with the occurrence of rim-runners. In addition to malpositioning, a difficult ligament situation was possible owing to the occurrence of rim-runners and higher inserts. This was confirmed by the correlation between the height of the inserts and the total wear area on the lateral compartments of the inserts. However, the data presented here revealed no correlation between wear patterns and areas and the implant size. The use of optical microscopy at low magnifications and simple grading systems [21,27] may improve retrieved inserts’ evaluation.

To describe the wear behavior of the total knee implants, patient-individual aspects may be disregarded. Patient-specific data were mainly reduced to the implantation time. Information about the reason for primary or revision surgery, body weight, activity level, age, gender, and other patient-specific parameters might be helpful in classifying the wear data.

## 5. Conclusions

In the past, differences and similarities between wear simulator tests and in vivo situations could be seen [19,20,21,25,26]. Comparison between studies was limited due to the comparison of different total knee designs [21], examination of only posterior-stabilized [20] or unicondylar designs [19], or focus on micro-wear patterns [26] or total wear areas [20,25]. Wear areas and patterns seem to be influenced by implant design and material type. Hence, despite the usage of the same classifications of damage and similar grading systems, direct comparison of study data is difficult. Therefore, the results of studies regarding wear area and wear pattern of retrieved and simulator-tested inserts can differ [22]. However, the data presented here from retrieved and simulator-tested UHMW-PE inserts using an identical total knee system show significant findings regarding the comparability of retrievals and simulator-tested inserts, as listed below:Deep wear zones could be better replicated according to the in vivo situation based on the third body wear simulation compared with other wear test conditions, despite similar in situ times of the retrieved and the simulator tested inserts (according to Battenberg et al. [31]).Pitting and TBW, the most prevalent wear patterns in the retrievals, could not be reproduced in the standard wear simulator test but could be replicated in the worst-case wear test conditions.Single rim-runners were detected on all the retrievals but could not be reproduced in the standard wear simulator tests; they were detected only in tibial sloped and internal rotated simulations.

The best accordance for the implant design was found between TBW-simulator-tested and retrieved inserts. A TBW-simulator test seems to be much more reasonable for preventing improper preclinical prediction of the wear behavior. By comparing the wear areas and wear patterns on the retrieved inserts with those on the simulator-tested inserts under different wear test conditions, preclinical testing may be adjusted to improve the prediction of the in situ performance of total knee implants in the future. Further, it may be possible to configure more physiological test scenarios, i.e., in line with younger and more active patients.

## Figures and Tables

**Figure 1 jfb-13-00256-f001:**
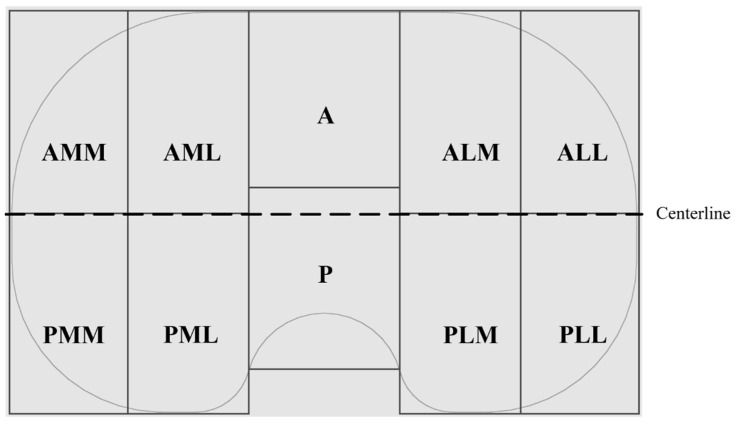
Schematic of the proximal surface of a right insert with partitioning in anterior (A) and posterior (P) portions, as well as medial (M) and lateral (L) compartments on each side.

**Figure 2 jfb-13-00256-f002:**
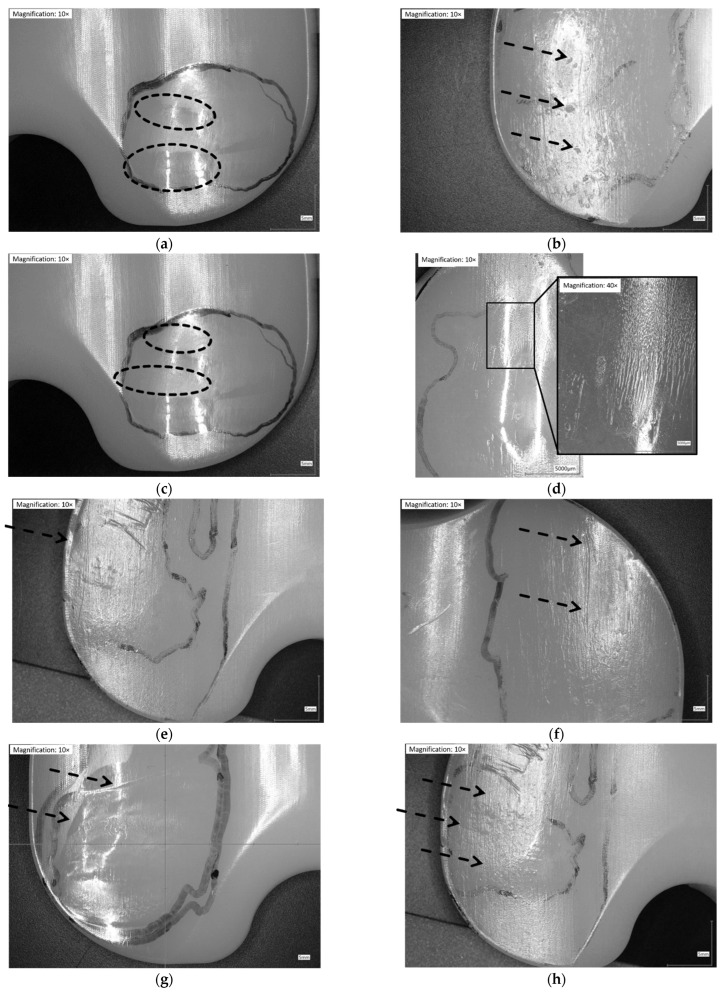
Illustration of wear patterns found on the retrieved UHMW-PE inserts. (**a**) Burnishing: polishing of the surface → surface appears smoother than manufacturing marks; (**b**) Pitting: depressions in articulating surface with irregular shape (2–3 mm across and 1–2 mm deep); (**c**) Abrasion: roughening of the surface, which results in a shredded or tufted appearance → surface appears rougher than manufacturing marks; (**d**) Striated wear pattern: described in a study [9] as regularly spaced striations in an anterior-posterior direction; (**e**) Rim-runner [24]: propagation of the worn areas on the elevated, rounded rim of the insert, with a localized form of creeping; (**f**) Third-body wear: parallel scratches causing thick scars of material removal (sometimes distinct holes from temporarily embedded debris at one end); (**g**) Deformation/Deep wear: distinct material accumulation on or around articulating areas; (**h**) Scratching: indented lines, generally in antero-posterior direction with only slight material removal.

**Figure 3 jfb-13-00256-f003:**
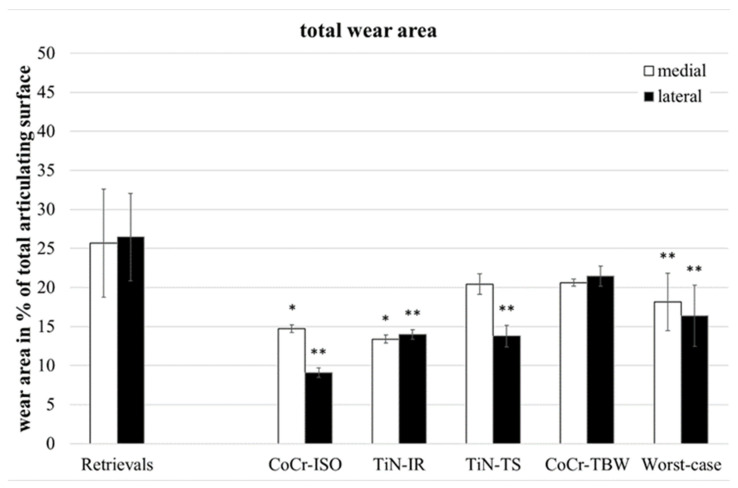
Distribution of percentual total wear areas. Significances, marked with asterisks (one asterisk *: *p* < 0.05 and two asterisks **: *p* < 0.01), are shown to facilitate the comparison of retrievals with simulator-tested groups.

**Figure 4 jfb-13-00256-f004:**
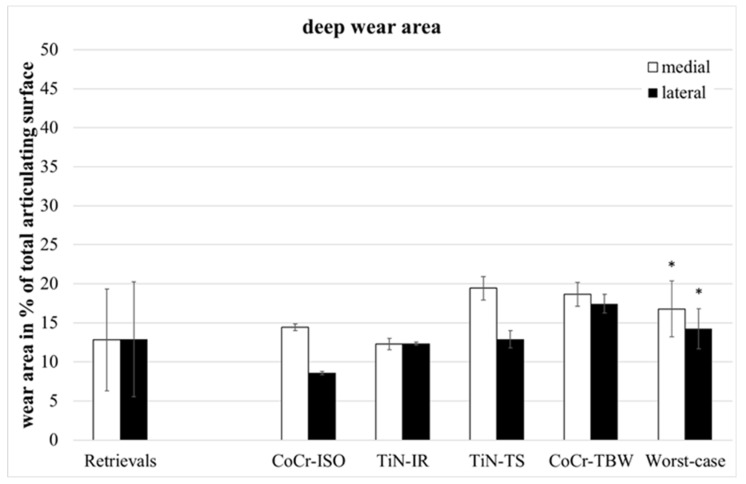
Distribution of percentual deep wear areas. Significances, marked with asterisks (one asterisk *: *p* < 0.05), are shown to facilitate the comparison of retrievals with simulator-tested groups.

**Figure 5 jfb-13-00256-f005:**
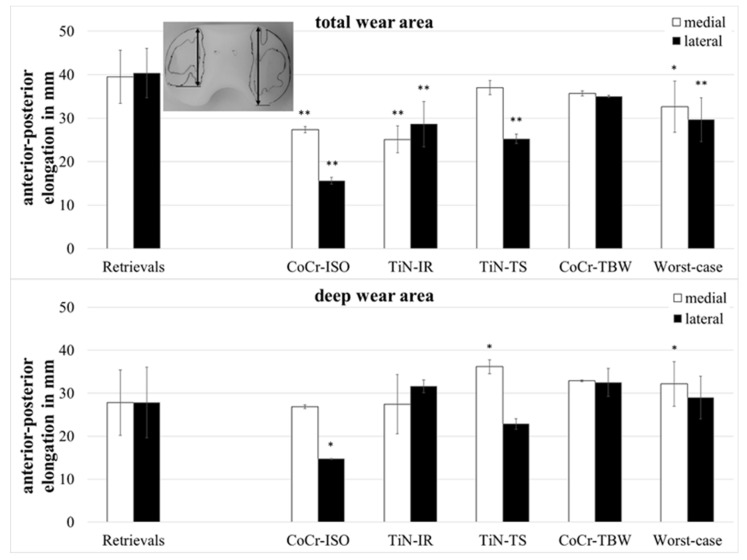
Dimensions of anterior–posterior elongation of wear areas of retrieved and simulator-tested UHMW-PE inserts. Significances, marked with asterisks (one asterisk *: *p* < 0.05 and two asterisks **: *p* < 0.01), are shown to facilitate the comparison of retrievals with the simulator-tested groups.

**Figure 6 jfb-13-00256-f006:**
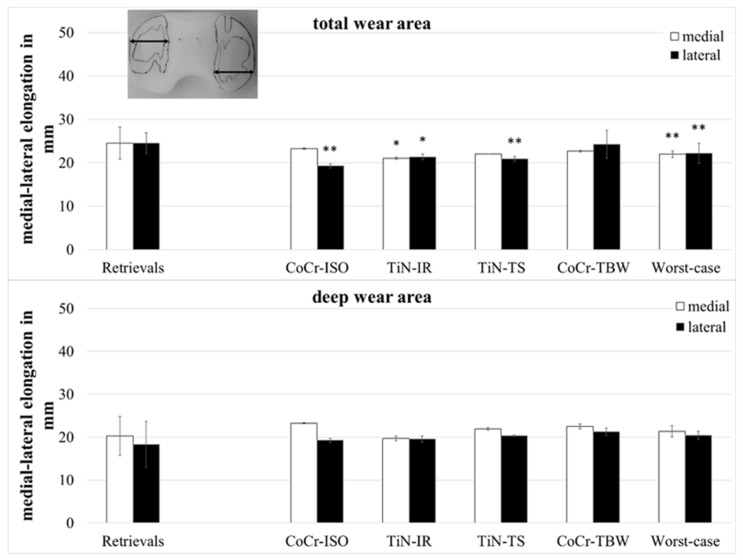
Dimensions of medial–lateral elongation of wear areas of retrieved and simulator-tested UHMW-PE inserts. Significances, marked with asterisks (one asterisk *: *p* < 0.05 and two asterisks **: *p* < 0.01), are shown to facilitate the comparison of retrievals with the simulator-tested groups.

**Figure 7 jfb-13-00256-f007:**
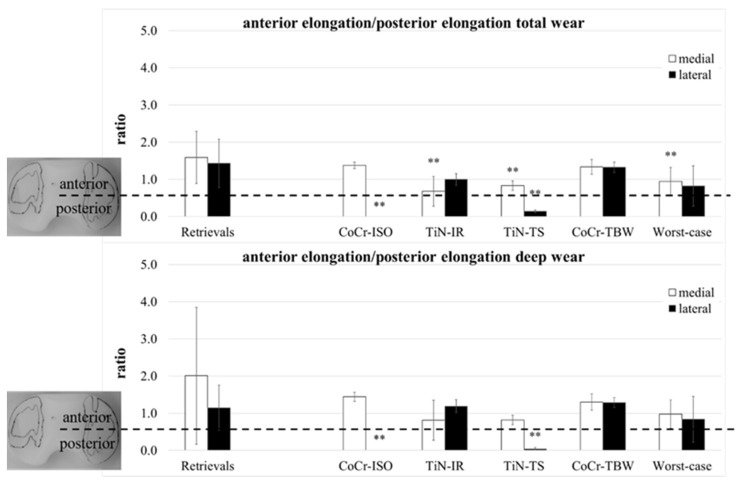
The ratio of anterior part of ap-elongation and posterior part of ap-elongation of retrieved and simulator-tested UHMW-PE inserts for total and deep wear areas. Significances, marked with asterisks (two asterisks **: *p* < 0.01), are shown to facilitate the comparison of retrievals with the simulator-tested groups.

**Figure 8 jfb-13-00256-f008:**
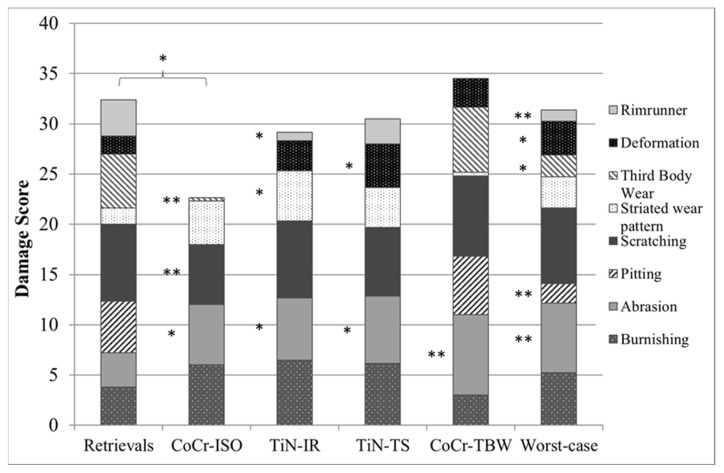
Distribution of each wear pattern on the damage score for retrieved and simulator-tested inserts. Significances, marked with asterisks (one asterisk *: *p* < 0.05 and two asterisks **: *p* < 0.01), are shown to facilitate the comparison of retrievals with the simulator-tested groups. (Maximum score for each wear pattern was 8, Maximum total damage score was 64).

**Table 1 jfb-13-00256-t001:** Tested conditions for the inserts in the simulator with the corresponding femoral components and abbreviations used in the further text.

Test Condition	Standard Wear Test, ISO 14243-1	Worst-Case Tests		
		Internal Rotation (IR) of Femoral Component of 9°	Posterior Tibial Slope (TS) of 10°	Third-Body-Wear (TBW)
Femoral component	Co28Cr6Mo (1)	Co28Cr6Mo and TiN (2) coating	Co28Cr6Mo and TiN (2) coating	Co28Cr6Mo (1)

(1) CoCr in the further text, (2) TiN = Titanium Nitride.

**Table 2 jfb-13-00256-t002:** Overview of all collected data for retrieved and wear simulator-tested inserts, values given as mean ± standard deviation.

	Time In Situ in Years	Total Wear in %	Superficial Wear in %	Deep Wear Area in %	Damage Score	Wear Pattern		
						1.	2.	Further
**Retrievals (*n* = 21)**	1.9 ± 1.9	52.3 ± 11.7	29.6 ± 10.3	22.6 ± 12.8	32.7 ± 5.9	scratching	pitting TBW	rim-runner, burnishing, abrasion
**CoCr-ISO (*n* = 3)**	2.5 (1)	23.9 ± 1.1	0.8 ± 0.9	24.6 ± 0.9	22.7 ± 1.5	burnishing, abrasion, scratching	striated wear pattern	-
**TiN-IR** **(*n* = 3)**	1.2 (1)	27.4 ± 1.0	2.8 ± 0.6	24.6 ± 0.9	29.2 ± 4.9	scratching	burnishing, abrasion	striated wear pattern,rim-runner
**TiN-TS** **(*n* = 3)**	1.5 (1)	34.3 ± 2.4	1.9 ± 0.8	32.4 ± 2.3	30.5 ± 3.0	burnishing, abrasion, scratching	striated wear pattern, deformation	rim-runner
**CoCr-TBW** **(*n* = 3)**	2.5 (1)	42.1 ± 1.1	6.0 ± 2.0	36.1 ± 1.6	35.0 ± 1.8	abrasion, scratching	TBW	pitting
**Worst-case** **(*n* = 12)**	1.7 ± 0.6 (1)	34.6 ± 6.5	3.6 ± 2.2	31.0 ± 5.3	32.0 ± 4.0	abrasion, scratching	burnishing	striated wear pattern, deformation

(1) According to Battenberg et al. [31].

## Data Availability

Not applicable.

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
