# Peer review of "A Comparison of Wear Patterns on Retrieved and Simulator-Tested Total Knee Replacements"

_jfb, 2022, doi:10.3390/jfb13040256_

Round 1
Reviewer 1 Report
The subject matter of this manuscript deals with the comparison of wear patterns between retrieved total knee replacements (TKR) from revision surgeries and simulator-tested TKR by examining wear area.
Seemingly, the manuscript has been well-written with the sufficient research outputs. However, it is very unfortunate that there are neither any data nor results related to ‘functional biomaterials’, e.g., synthesis, fabrication, and surface modification or functionalization. Thus, it is recommended that the authors submit this paper to a specific or related clinical journal more suitable for its scope.
Reviewer 2 Report
Comments:
1. The abstract should state briefly the purpose of the research, the principal results and major conclusions. Please revise the abstract.
2. The introduction is short and should be expanded.
3. The introduction should be dedicated to present a critical analysis of state-of-the-art related work to justify the objective of the study. In this case, overall, the introduction section is too simple, without a comprehensive description
4. The purpose and significance of this study should be explained in the introduction
5. The authors should highlight the significance/novelty of their work.
6. Please improve the presentation of all Figures in terms of quality and clarity
7. Kind suggestions, clearly demonstrate the aims or objectives of this work.
8. The authors should explain which exact problem could be solved by the present research.
9. The author needs further explanation of the uniqueness of this work in comparison to the existing works done with the previous reports
10. To strengthen the discussion and justify the results, authors are suggested to go through some recent and very important reference papers and must include them in the revised manuscript.
11. Please discuss further the fact that the method is not entirely successful, identifying the origin of the problem and suggesting further improvements as potential future work.
12. The conclusions section should be re-written. Please clearly mention the most important results and outcomes of your work.
13. Social implications shall be highlighted in the conclusion
14. Authors are recommended to revise the whole manuscript for the language proof
Reviewer 3 Report
Dear Authors,
Thank you for very interesting paper considering wear of knee replacements. I think that the obtained results will have a significant impact on the development of the biomedical field.
I have some suggestions, which could improve reading of paper.
1. The measurable values in the charts should be added.
2. What do the asterisks in the figure 8 mean?
3. Why the time in situ for retrievals has a deviation (Table 2)?
Best regards,
Reviewer
Round 2
Reviewer 1 Report
The authors provided replies to some critical issues raised in the previous review stage and the present version of the manuscript was revised to some extent.
Although the authors responded that their manuscript topic is congruent with this SI, it is considered that this paper is more suitable for a specific or related clinical journal.
Reviewer 2 Report
The revised version of the manuscript is good enough to be published. The authors addressed each comment in detail and made appropriate changes.